# Experimental Investigation of the Evaluation of the Cement Hydration Process in the Annular Space Using Distributed Fiber Optic Temperature Sensing

**DOI:** 10.3390/s25030958

**Published:** 2025-02-05

**Authors:** Zhong Li, Mengbo Li, Huan Guo, Yi Wu, Leixiang Sheng, Jingang Jiao, Zhenbo Li, Weibo Sui

**Affiliations:** 1CNOOC Res Inst Co., Ltd., Beijing 100028, China; lizhong@cnooc.com.cn (Z.L.); limb7@cnooc.com.cn (M.L.); lizhb20@cnooc.com.cn (Z.L.); 2College of Petroleum Engineering, China University of Petroleum, Beijing 102249, China; 2024310190@student.cup.edu.cn

**Keywords:** cement hydration process, distributed fiber optic sensing (DFOS), cement sheath integrity, wellbore integrity

## Abstract

This study employed a full-scale cement sheath quality evaluation apparatus, along with a high-precision distributed fiber optic temperature sensing system, to perform real-time, continuous monitoring of the temperature change throughout the cement hydration process. The results of the cement annulus and cement bond defect monitoring during the hydration process indicated that the distributed fiber optic temperature data enabled centimeter-level resolution in defect identification. Defective regions exhibited significantly reduced temperature fluctuation amplitudes, and an inversion in temperature change at the early hydration stage, detected at the cement–defect boundary, facilitated the early detection of defect locations. The distributed fiber optic system was capable of conducting continuous and comprehensive monitoring of the sequential hydration temperature peaks of cement stages injected into the annulus. The results revealed the interdependence among different cement stages, as well as a phenomenon whereby an elevated annular temperature accelerates the progression of cement hydration. The experimental findings provide a reference for identifying the characteristic signals in distributed fiber optic monitoring of well-cementing operations, thereby establishing a foundation for the optimal and effective use of distributed fiber optics in assessing well-cementing quality.

## 1. Introduction

Well cementing creates a secure barrier between the casing and the formation, playing a critical role in maintaining the structural integrity of wellbores and ensuring the effective use of the casing throughout the entire life cycle of oil and gas wells [1]. During drilling and hydraulic fracturing, high-volume fluid is continuously injected into the casing, subjecting both the wellbore and the annular cement to significant pressure. With repeated fracturing, cyclic exposure to extreme temperatures and pressures induces irreversible plastic deformation in the cement sheath, compromising its integrity [2]. Once the integrity of the cement sheath is compromised, high-pressure formation fluids can easily infiltrate the cement sheath during subsequent production stages. This infiltration further reduces the bond strength between the cement sheath, formation, and casing, resulting in severe sustained casing pressure issues and posing a significant threat to the operational safety of oil and gas wells [3]. The strength of the cement sheath gradually increases throughout the hydration process, and the accurate monitoring of temperature and pressure logging data during cement hydration enables the estimation of the post-hydration strength and the integrity of the cement sheath, providing crucial insights for planning and executing subsequent well completion and production operations. However, monitoring the downhole quality of well-cementing operations has long been a challenging task. Traditional monitoring methods, such as downhole temperature and pressure sensors, exhibit poor adaptability to harsh conditions, often failing to reach the precision required for such measurements. Consequently, these methods face significant limitations in identifying specific downhole events during cementing and in supporting long-term monitoring [4]. Acoustic, ultrasonic, and amplitude monitoring techniques have relatively low spatial resolution along the measured depth profile, and their measurement accuracy is constrained. When affected by complex topography and formation conditions, it becomes challenging for these techniques to capture accurate and effective signals for analysis [5,6]. Furthermore, traditional well-cementing logging techniques can only perform monitoring at single or multiple points during specific time intervals, making it impossible to achieve real-time, long-term measurements across the entire length of the cemented section.

Distributed fiber optic sensing technology offers powerful support and novel possibilities for real-time, continuous monitoring and analysis of well cementing. By deploying fiber optic cables inside and outside the casing, real-time, continuous monitoring of the wellbore temperature and strain fields can be conducted during both the cementing process and subsequent production phases. Fiber optics inherently possess excellent properties, such as electromagnetic interference resistance, high-temperature tolerance, and corrosion resistance, which enable them to adapt to extreme downhole environments while maintaining measurement accuracy [7,8]. By using distributed temperature sensing (DTS) technology to monitor anomalous temperature response regions in the cement sheath and by combining these data with cementing operations and the hydration time, the degree of cement hydration can be assessed, the cement loss zones and top depth can be identified, and the quality of the cementing operation can be evaluated [9].

Ricard [10] utilized DTS outside the casing to monitor the cementing process of a vertical well and observed significant anomalous high-temperature occurrences at multiple locations in the formation, caused by the accumulation of cement due to drilling-induced losses. Additionally, subsequent DTS monitoring during borehole heating tests revealed uneven heat transfer along the depth of the wellbore, further confirming the influence of cement thickness heterogeneity on temperature distribution. Lee [11] utilized DTS to monitor the cooling of cement slurry circulation and the subsequent temperature rise during the cement hydration process in a vertical CO_2_ sequestration well. DTS temperature data indicated that, at the onset of cement hydration, temperature peaks were observed at both the top and bottom of the cement column. As hydration progressed, these temperature peaks gradually propagated toward the center of the cement column. Lipus [12] employed DTS installed outside the production casing to monitor the annular flow rate of cement slurry in a vertical geothermal well, which was used to determine the displacement efficiency of the slurry. Additionally, DTS monitoring captured complex downhole conditions, such as cement slurry infiltration into the formation and dilution by formation fluids entering the annulus. Wu [13] utilized a high-precision fiber optic temperature monitoring system in the laboratory to observe temperature during the cement hydration process, verifying the sensitivity of fiber optics to temperature variations under conditions of drilling fluid dilution and annular defects, as well as the feasibility of such monitoring. Chen [14] conducted a study using a fiber Bragg grating to monitor the strain response at the cement interface under temperature and pressure cycling conditions in a full-scale cement sheath, providing a comprehensive description of the cement sheath failure process. However, the discontinuity of fiber Bragg gratings under complex and dynamic downhole conditions may limit their monitoring effectiveness to some extent. Achieving distributed fiber optic monitoring of full-scale cement sheaths under complex borehole conditions in the laboratory is of significant importance for elucidating the mechanisms of fiber optic signal response and aligning these insights with oil field well-cementing fiber optic monitoring operations, thereby fully leveraging the advantages of fiber optic monitoring.

In this study, a high-precision distributed fiber optic strain and temperature measurement system will be used to monitor the cement sheath and bond interface during the hydration process under defective conditions. The distributed fiber optic system offers enhanced resolution while maintaining measurement accuracy, enabling the precise identification and localization of anomalies within the cement sheath during hydration. Unlike previous laboratory-based monitoring methods, this study conducts cement sheath hydration monitoring based on the annular space, which is defined by the combination of casing dimensions under field conditions. This allows for a more accurate representation of the data variation amplitude, closely aligning with actual field conditions. A wellbore length of 1 m effectively differentiates signal anomalies caused by sensor instability or interference from cement non-homogeneity from those due to the absence of the cement sheath or defects in the bonding interface. The analysis of the temperature response characteristics of the cement sheath will provide a reference for interpreting field well-cementing fiber optic data.

## 2. Experiment Setup and Methodology

### 2.1. Experiment Apparatus and Materials

#### 2.1.1. High-Precision Distributed Fiber Optic Temperature/Strain Measurement System

This experiment employed a dynamic distributed fiber optic sensing system based on optical frequency domain reflectometry (OFDR) to monitor temperature in real time throughout the entire experimental process (Figure 1). As shown in Figure 1, the fiber optic monitoring system primarily consists of an interrogator (OSI-D), monitoring software installed on a computer connected to the interrogator, which allows for the configuration of monitoring parameters such as spatial resolution, monitoring mode (temperature/strain), and data storage, as well as a single-mode optical fiber for data sensing and transmission. The OSI–D system achieves a spatial resolution of up to 0.64 mm, with a temperature measurement accuracy of ±0.1 °C and a strain measurement accuracy of ±1 με, at a sampling rate of 120 Hz. The specific equipment parameters are provided in Table 1 [15].

Through optimized algorithms, the real-time dynamic demodulation of the OFDR signals is achieved. OFDR is a coherent detection technique based on continuous optical frequency modulation. It determines the location of scattering signals by measuring the frequency of Rayleigh backscatter generated by the modulated probe light, offering millimeter-level spatial resolution and exceptionally high sensing accuracy. The basic configuration of OFDR is illustrated in Figure 2. The linearly swept light emitted by a tunable laser is split into two paths by an optical coupler. One path enters the reference fiber, where it is reflected back by an end mirror reflector to serve as the local oscillator reference light. The other path enters the sensing fiber, where it detects temperature or strain variations acting on the fiber. Rayleigh backscatter is continuously generated within the sensing fiber, propagating back along the original path. The two backscattered light beams undergo beat interference in the coupler before entering the photodetector, where the optical signals are converted into electrical signals. This process yields the Rayleigh backscatter distribution along the entire fiber. By demodulating the Rayleigh backscatter spectral signals, strain and temperature data can be obtained [16,17].

#### 2.1.2. Full-Scale Cement Sheath Sealing Performance Evaluation Apparatus

The full-scale cement sheath sealing performance evaluation apparatus consists of a wellbore physical simulation system and a data acquisition system. As shown in Figure 3, the wellbore system comprises the outer casing, inner casing, and cement annulus. The inner casing has an outer diameter of 139.7 mm and a wall thickness of 7.72 mm and is made of commonly used oilfield casing material with a grade of P110. The cement annulus has a thickness of 31.65 mm, and the outer casing is made of steel grade N80. Both the inner and outer casings have a length of 1000 mm. The high-precision distributed fiber optic temperature and strain measurement system serves as the data acquisition system. In Figure 4, the single-mode optical fiber is helically wound around the outside of the inner casing and connected to the interrogator. The OSI-D interrogator measures the Raleigh frequency shift during the experimental process through a single-mode optical fiber, with the data stored on the computer connected to the OSI-D. The temperature changes monitored by the thermocouple and their correlation with the Rayleigh frequency shift are extracted, and the Rayleigh spectrum is converted into a temperature change spectrum. We utilized Python to read the data, performing necessary downsampling and denoising procedures to obtain the spectrograms.

#### 2.1.3. Experiment Materials

Class G cement was selected for the experiment according to GB/T10238—2015, with the formulation consisting of 800 g of cement, 2% G33S fluid loss additive, 4% SCFP–H elastic material, 42% water, and 1% defoamer. Foam with a certain strength was used in the experiment to simulate conditions of annular cement loss and primary cement bond loss. Thermocouples were installed at specific locations to monitor the temperatures of both the annular cement and foam regions, serving to calibrate the fiber optic measurement results.

### 2.2. Experiment Design

In the experimental process, the fiber optic was installed on the exterior of the inner casing, which effectively simulates field monitoring conditions. Given the limitations of the laboratory scale, simply attaching the fiber optic along the axial direction of the casing would only reflect temperature changes along that axis. Furthermore, insufficient contact between the fiber and the casing could compromise measurement accuracy or even lead to data loss. By helically wrapping the fiber optic around the casing, the measurement length is effectively increased, thereby enhancing the tolerance of the fiber installation. Moreover, this configuration allows for multidirectional measurements, which facilitate the acquisition of more comprehensive and valid experimental data under laboratory conditions. In this experiment, the fiber optic was helically wrapped around the casing at an angle of 20°. An optimal winding angle is crucial for ensuring both precise data measurement and a sufficiently long optical fiber measurement length. Experimental trials with smaller angles were conducted; however, the noise induced by light loss from fiber bending was found to be unacceptable. Consequently, a winding angle of 20° was selected for this study. The experimental content covers three main aspects, with the design principles being closely aligned with oil field well-cementing issues.

#### 2.2.1. Temperature Change Monitoring of Cement Hydration Under Annular Deficiency

There are various causes of cement sheath deficiency in the field. Factors such as formation losses caused by drilling, high formation permeability, and formation fracture growth can lead to cement slurry losses and accumulation at specific locations. Inadequate cementing design and operations can cause channeling in the annulus, preventing the complete sealing of the annular space. Severe casing eccentricities, particularly in deviated sections of horizontal wells, result in uneven annular gaps. In narrower gaps, poor displacement efficiency leaves behind drilling fluids, mud cakes, and cuttings, causing incomplete cementing (Figure 5a). The exothermic hydration of cement slurry can disrupt the thermal and pressure equilibrium of hydrate formations, causing gas release that infiltrates and accumulates in the cement, resulting in annular deficiencies [18,19,20,21]. As shown in Figure 5b,c, foam was used to simulate annular cement deficiencies, and fiber optic temperature responses were studied in both foam and cement segments. Foam with different thicknesses (20 mm and 60 mm) was used to assess the sensitivity of fiber optic temperature signals to deficiency size. In the experiment setup, the fiber optic was helically installed and secured around the inner casing outside, and its connection with the equipment was verified, ensuring an optical loss within 35 dB. Thermocouple wires were positioned at the predetermined cement and foam locations with appropriate markings. Subsequently, the inner and outer casing were assembled, cement was poured into the annulus to the designed height, and annular foam of specified thickness was placed. The annulus was then sealed for hydration monitoring.

#### 2.2.2. Temperature Change Monitoring of Cement Hydration Under Bond Interface Deficiency

Factors such as drilling fluid and mud cake contamination, cement shrinkage during hydration, fluid loss, and gas migration before cementation can cause poor bonding between the cement and casing, leading to microcracks that affect cementing quality [22] (Figure 6a). In the experiment, foam with heights of 60 mm and widths of 10 mm and 5 mm was attached to the casing exterior to simulate bond interface deficiencies (Figure 6b,c). This was done to verify the feasibility of using DTS to identify bond deficiencies during the hydration process, as well as to evaluate sensitivity to different deficiency sizes.

#### 2.2.3. Temperature Change Monitoring of Annular Segmented Cement Hydration

As mentioned, due to issues like formation loss during drilling or intersections with faults, significant cement slurry loss often occurs during well cementing. Therefore, staged cementing techniques are typically employed in the field to mitigate these losses. For ultra-deep wells, the success rate of single-stage cement return and cementing quality is often low, necessitating the use of stage collars and packer cementing tools to perform staged cementing [23,24,25] (Figure 7a). Distributed fiber optics enable the continuous monitoring of the staged cementing process, reducing the cost and risks associated with re-deploying cement logging equipment. In this experiment, subsequent segments of cement were injected at one-day intervals based on the annular defect experiment. The experimental procedure involved injecting the first segment of cement on day one (Figure 7b), followed by hydration for one day, then injecting the second segment, and finally the third segment after another day, sealing the annulus for hydration monitoring (Figure 7c,d). As the cement hydration assessment was conducted at room temperature, a one-day interval between injections ensured distinct hydration progress for each segment. This experiment also simulated the effect of geothermal gradients, which cause variations in hydration progress across different formation depths. Although the experiment did not directly simulate temperature variations in the formation environment, it captured the essence of how different environmental temperatures affect hydration progress, with different segments experiencing varied hydration processes. The distributed fiber optic temperature responses under both conditions are similar.

### 2.3. Experimental Data Processing

Figure 8 illustrates the positioning of three thermocouples at the midpoint within the cement section and the deficient sections. The fiber optic response signals (Rayleigh frequency shift), placed at the same locations as the thermocouples, are compared with the temperature variations recorded by the thermocouples. During the experimental monitoring process, noise and oscillations in the data were inevitable due to equipment stability and environmental fluctuations. To facilitate data analysis and computation, denoising and smoothing were applied to the experimental data.

Figure 9 shows the processed fiber optic response and thermocouple measurements from the annular cement defect hydration monitoring experiment. The trends in fiber optic response at the cement and foam locations strongly correlate with the temperature change trends recorded by the thermocouples at the corresponding positions.

Additionally, it should be noted that the fiber optic response is simultaneously influenced by both temperature and strain. In this study, the hydration of the cement results in its solidification and adhesion to the fiber, while the cement’s autogenous shrinkage may induce strain on the fiber within the cement sections. A comparison of the fiber response curves measured within the cement and foam sections is provided in Figure 9. This comparison revealed that both types of fiber responses followed the same trends as the thermocouple temperature measurements. It suggests that the strain interference is negligible and will not significantly impact the temperature monitoring outcomes.

A polynomial fitting method was used to establish the relationship between fiber optic response and thermocouple temperature changes at the same measurement point. The fiber optic response signals at the three points shown in Figure 9 were fitted to the corresponding thermocouple temperature changes. The left axis represents the values of the dashed line, and the right axis represents the values of the solid line. A polynomial with a low mean squared error (MSE) was then used to estimate the temperature changes at other measurement points within the cement and foam sections, yielding the distributed temperature change results. Figure 10 presents the fitting results for the three points, showing that the MSE for cement and 20 mm foam against the thermocouple temperature changes is approximately 0.35 °C. The MSE at the 60 mm foam location is larger at 2.05 °C. This may be due to the relatively lower temperature at the thick foam section and the greater influence of cement shrinkage strain, which reduces the fiber optic temperature response. A third-degree polynomial was used for fitting. Although increasing the polynomial order could reduce the MSE, it also induces data oscillations; therefore, a further reduction in the MSE was not pursued. The polynomial with the lowest MSE was selected to fit the temperature changes for the remaining fiber optic measurement points. The subsequent analysis focuses on the temperature differences between the cement and foam sections, and the fitting error mentioned above does not affect the relative relationship between these sections.

## 3. Experimental Results and Analysis

### 3.1. Analysis of the Temperature Change for Cement Hydration Under Annular Deficiency

Figure 11 shows the temperature changes during the hydration process of the annular cement deficiency. The left side illustrates the depth of the deficiency along the axial direction of the inner casing and the thickness of the deficiency. The right side of the figure shows the temperature change results obtained from fiber optic monitoring within the selected region. Overall, the fiber optic captured the non-uniform temperature changes, indicated by discontinuous horizontal bands, which are correlated with factors such as cement quantity, type, ambient temperature, and injection procedure. During the first 10 h, the temperature increased slowly, followed by a rapid rise from 10 to 22 h as significant heat was released by the cement, peaking near 20 °C. After the peak, the temperature decreased at a slower rate. The white dashed lines indicate the boundaries between the deficiency and surrounding cement. The temperature change in the thicker deficiency is significantly lower than that of the surrounding cement, while the thinner deficiency shows similar temperature changes to the cement.

Figure 12 shows the temperature changes at the midpoint within the cement and deficiency sections (indicated by the green triangle in Figure 11). Cement exhibits the largest temperature change, followed by the thin deficiency, with the thick deficiency showing the least. Foam sections receive heat transfer from the surrounding cement, resulting in a lower temperature change compared to the cement, which releases heat internally. The distinct difference between the thick and thin deficiency temperature changes demonstrates the fiber optic’s ability to distinguish annular deficiency thickness with a centimeter-level accuracy under laboratory conditions.

Figure 13 presents temperature changes along the fiber at 3, 22, and 50 h of the hydration process. The deficiency positions consistently exhibit lower temperature changes compared to the central cement section, with the thick deficiency being more prominent in the two green-dashed-line zones. At 22 h, the cement temperature reached its peak, and a distinct signal inversion was observed at the deficiency–cement boundary (elliptical region), contrasting with the minor fluctuations in the central cement area (rectangular region). The temperature inversion at the interface is due to the different thermal conductivity coefficients of foam and cement. Since the thermal conductivity coefficient of foam is lower, compared with the foam section, the cement section has a faster and larger temperature increase due to the hydration effect. However, heat transfer from the cement section to the foam section was hindered to some extent, thus the temperature near the cement–foam interface is higher than the middle section of the cement. Similar inversions also appeared at the boundary after 3 h of hydration, indicating that this signal characteristic can be used for the early prediction of potential annular deficiencies. Subsequent monitoring should focus on the temperature changes near such signal features. After 50 h of hydration, the temperature variations in the cement section reflected a cement heterogeneity, while the deficiency signal characteristics remained distinct. Late-stage temperature signals further corroborated the early and mid-hydration observations of the deficiency segments.

### 3.2. Analysis of the Temperature Change for Cement Hydration Under a Bond Interface Deficiency

Figure 14 shows the temperature changes during cement hydration under a bond interface deficiency. The left side represents the depth and location of the design defect, as well as the size of the defect. In this experiment, the width of the defect was modified. Similar to the annular deficiency scenario, the temperature peaked at 25 h but with a lower peak value of approximately 15 °C, likely due to the reduced insulation caused by poor sealing in this experiment. Similarly, the temperature change at the wider deficiency location was lower than that of the cement, while the narrower foam exhibited temperature changes closer to the cement.

Figure 15 shows the temperature changes at the midpoint within the cement and deficiency sections (indicated by the green triangle in Figure 14). Figure 15 further illustrates the relative temperature changes at the cement and deficiency positions, with cement being the highest, followed by narrow foam, with wide foam the lowest. The fiber optic temperature changes can distinguish the temperature differences between foams of varying widths. However, compared to the annular deficiency experiment, the differences are reduced, indicating the decreased sensitivity of fiber optic temperature monitoring to smaller-scale bond deficiencies.

As shown in Figure 16, the temperature variation at the deficiency location is smaller than that in the middle cement section at different time points in the two green-dashed-line zones. In contrast to the annular cement deficiency, the temperature drop at the deficient location is attenuated, and the temperature inversion at the foam–cement interface, induced by the thermal conductivity coefficient disparity between cement and foam, is also alleviated. This effect is directly correlated with the foam volume. The results indicate that fiber optic temperature monitoring can identify smaller-scale bond interface deficiencies, but the response characteristics are weaker. These deficiencies can be further confirmed by monitoring strain anomalies in the annular cement caused by subsequent high-pressure fluid extrusion.

### 3.3. Analysis of the Temperature Change for Annular Segmented Cement Hydration

Figure 17 shows the temperature change during the segmented cementing process; the interface depths of the different cement stages are marked on the left. The three cement stages were injected sequentially from bottom to top. Overall, the fiber optic monitoring clearly identified each cement stage, with the temperature peaks of each section occurring over time. The heterogeneity of the cement caused more distinct horizontal strip-like response signals compared to the previous experiments.

Figure 18 further illustrates the temperature variations during the hydration process for the three cement stages. The point located at the position of the green triangle in Figure 17 is selected. Upon the completion of the first cement injection (0 h), the unfilled upper annulus exhibited a temperature decrease due to the influence of the environmental temperature compared to the initial temperature at the start of the monitoring. After the second cement injection, the hydration of the second stage caused a temperature increase in the unfilled remaining annular space (ellipse region red line). Early in the second cement hydration, the temperature of the first cement stage remained higher than that of the second. Once the second stage reached its hydration temperature peak, the temperature decrease trend of the first stage leveled off. After the third cement section reached its peak temperature, the second section also showed a noticeable temperature increase (rectangular region blue line). In the later stages of the hydration process, the influence of environmental temperature (day and night) became more pronounced, causing the temperature change to exhibit a wave-like pattern.

Figure 19 compares the temperature changes of the three cement stages from injection to 48 h of hydration. The first stage showed a slow temperature rise initially at room temperature, while the second and third stages experienced accelerated early temperature increases due to the heat released by the previous cement sections. The time required to reach the peak hydration temperature decreased for each successive stage: 26 h for the first, 21 h for the second, and 20 h for the third. This indicates that the heat generated by the preceding stages accelerated the hydration process of the subsequent cement stages, reflecting the impact of the formation temperature at different measured depths on the cement hydration process.

## 4. Conclusions

(1) Based on a full-scale cement sheath fiber optic monitoring system, distributed fiber optic temperature monitoring was conducted to identify annular cement/cement bond deficiencies and sequential cement injections during hydration. The fiber optic results closely matched the thermocouple data, confirming the feasibility and accuracy of fiber optic monitoring for the cement hydration process.

(2) In the experiment on annular cement/cement bond deficiencies during hydration, distributed fiber optic temperature monitoring effectively distinguished between the cement and foam sections, with a defect recognition accuracy at the centimeter level under laboratory conditions. The deficiency sections showed significantly lower temperature changes compared to the cement, with clear temperature inversion signals at the cement–deficiency boundaries. These temperature responses were closely related to the defect volume. Larger defects caused inversion signals at the cement–deficiency boundary during early hydration, aiding the early detection of potential cement loss.

(3) In the experiment with sequential cement injections and hydration monitoring, the fiber optic system continuously tracked the hydration temperature peak of each cement stage as it reached its respective hydration phase. The fiber optic temperature effectively reflected the interactions between cement stages and the influence of environmental temperature on the cement hydration process, enabling the identification of accelerated hydration due to temperature variations in the annular space.

## Figures and Tables

**Figure 1 sensors-25-00958-f001:**
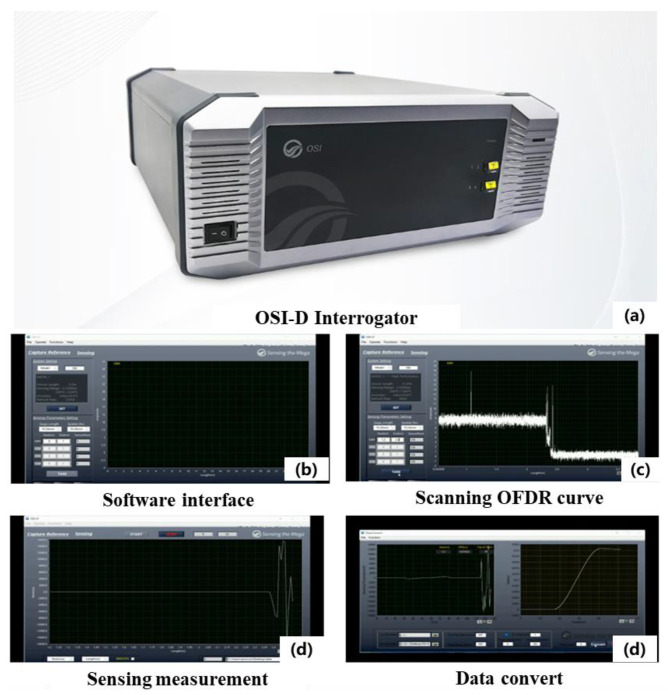
Dynamic distributed fiber optic sensing system.

**Figure 2 sensors-25-00958-f002:**
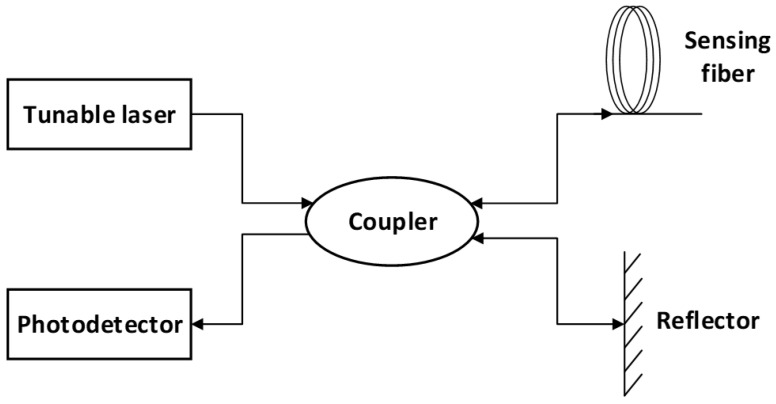
System architecture of OFDR.

**Figure 3 sensors-25-00958-f003:**
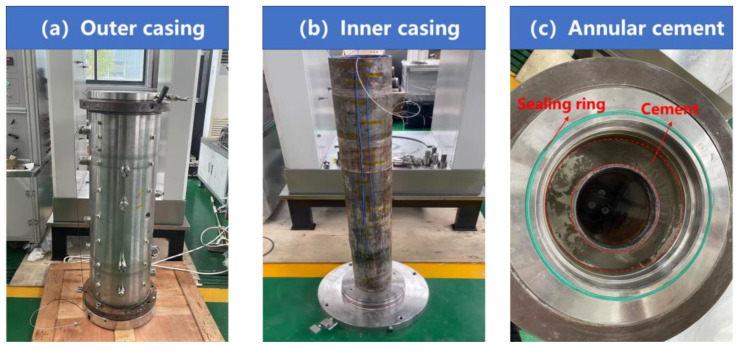
Wellbore physical simulation system.

**Figure 4 sensors-25-00958-f004:**
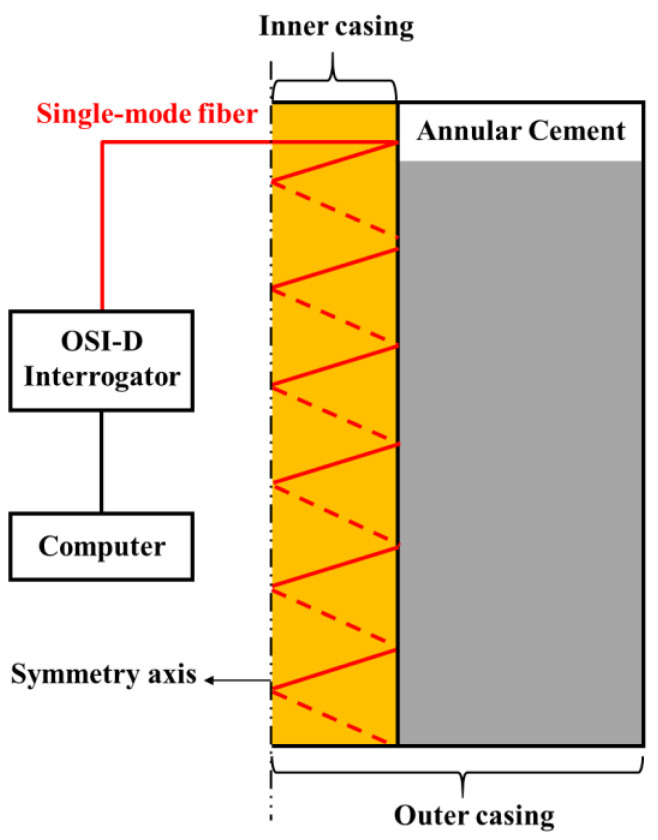
Schematic of optical fiber monitoring of cement hydration.

**Figure 5 sensors-25-00958-f005:**
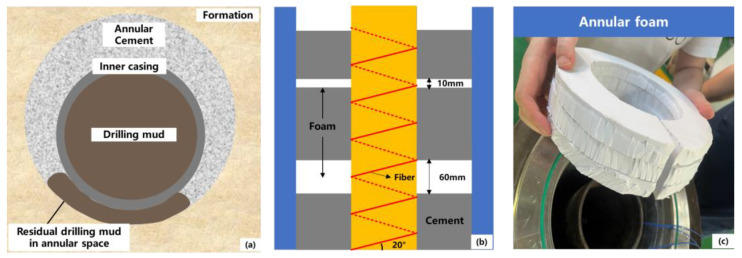
Experimental design for annular cement deficiency. (**a**) cement deficiency caused by residual drilling mud (**b**) design of annular deficiency with heights of 60 mm and 20 mm (**c**) foam used in simulating annular deficiency.

**Figure 6 sensors-25-00958-f006:**
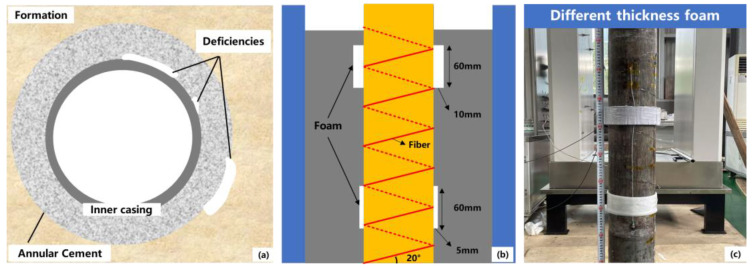
Experimental design for bond interface deficiency. (**a**) bond interface deficiency schematic (**b**) design of bond interface deficiency with widths of 10 mm and 20 mm (**c**) foam used in simulating bond interface deficiency.

**Figure 7 sensors-25-00958-f007:**
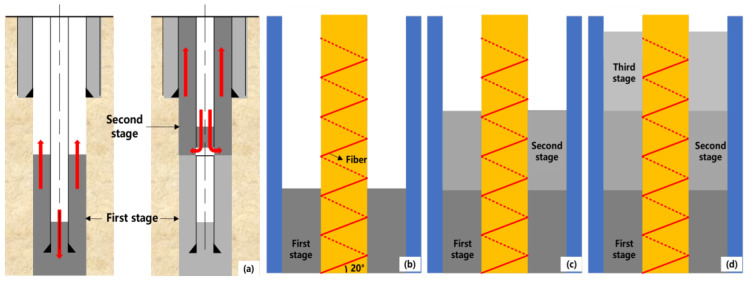
Experimental design for the assessment of annular segmented cement hydration. (**a**) the schematic of staged cementing, the experimental design of (**b**) the 1st (**c**) the 2nd (**d**) the 3rd stage of cementing.

**Figure 8 sensors-25-00958-f008:**
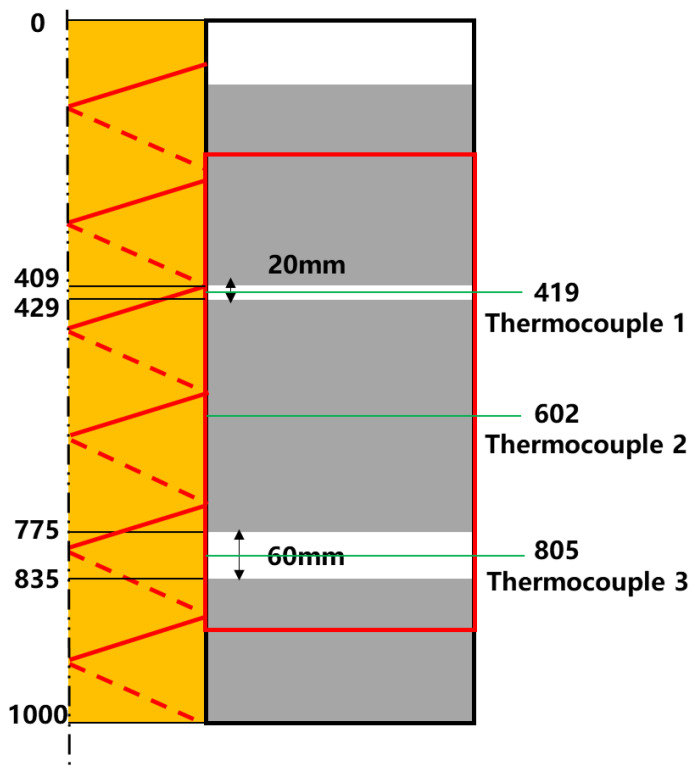
Schematic of the relative position of the thermocouple and optical fiber.

**Figure 9 sensors-25-00958-f009:**
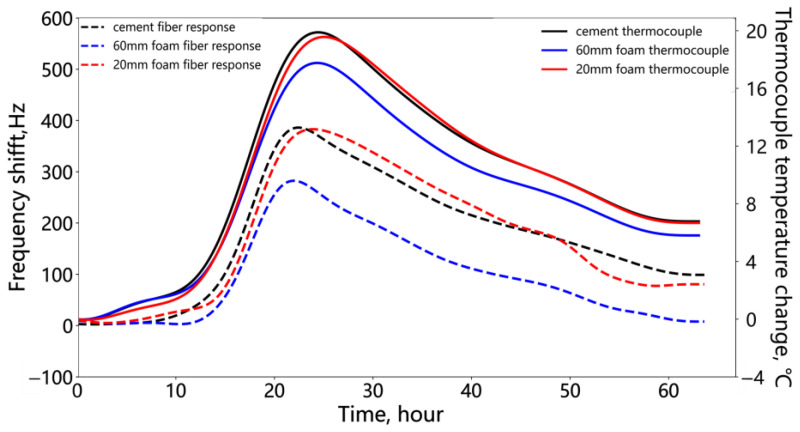
Fiber frequency shift and thermocouple temperature change during the hydration process of the annular cement deficiency.

**Figure 10 sensors-25-00958-f010:**
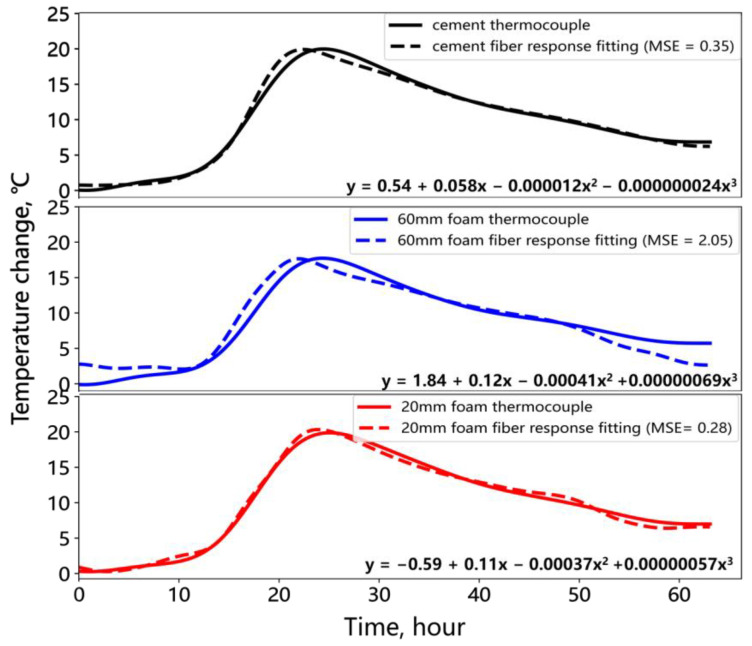
Fitting fiber optic response signals to temperature change.

**Figure 11 sensors-25-00958-f011:**
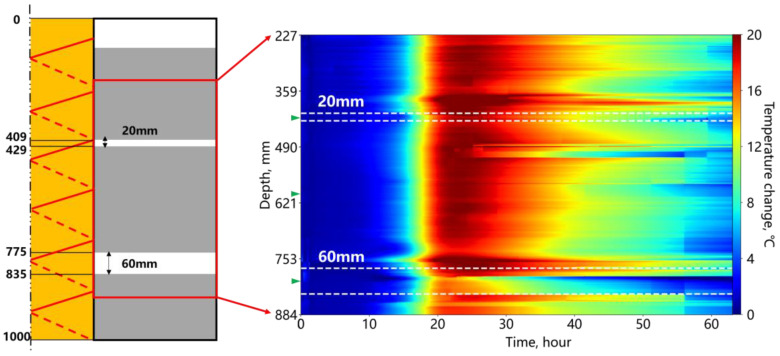
Temperature change during cement hydration in the annular deficiency.

**Figure 12 sensors-25-00958-f012:**
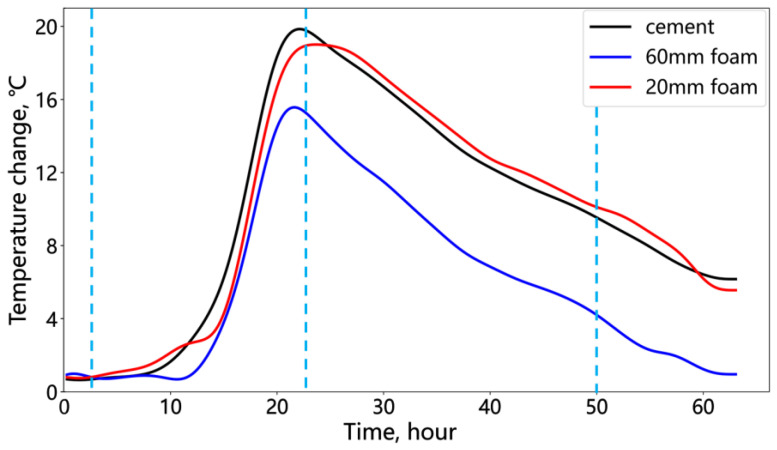
Temperature change of the cement and foam positions in the annular deficiency.

**Figure 13 sensors-25-00958-f013:**
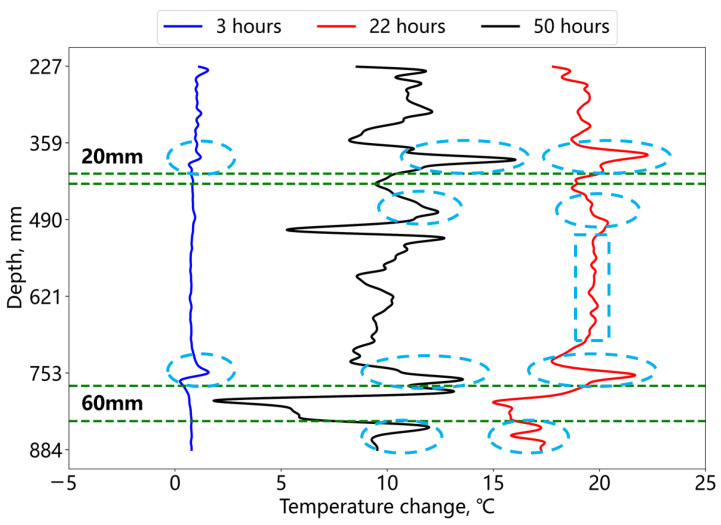
Temperature change at different hydration times for the annular cement deficiency.

**Figure 14 sensors-25-00958-f014:**
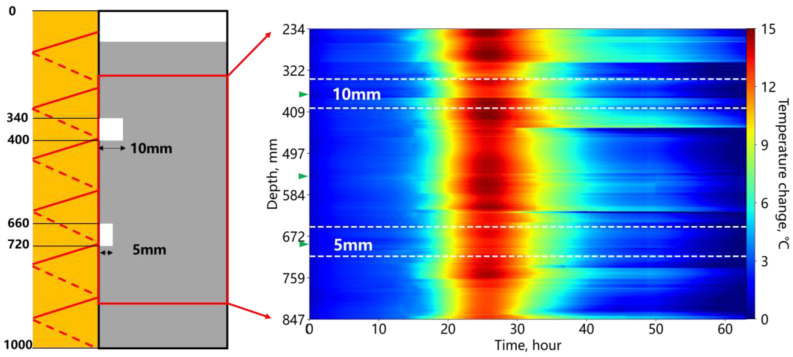
Temperature change due to the cement-bonding interface deficiency.

**Figure 15 sensors-25-00958-f015:**
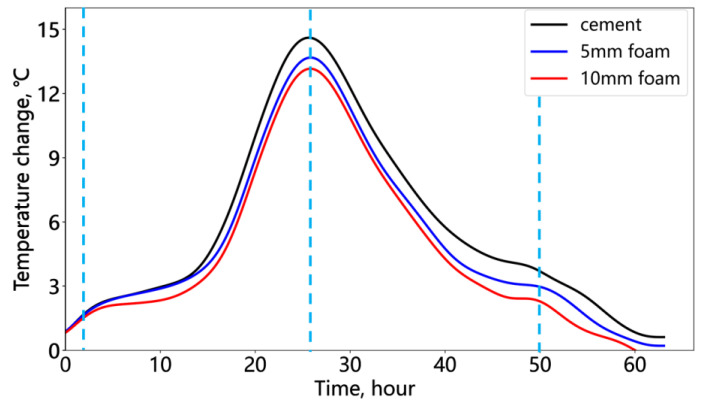
Temperature change of the cement and foam positions at the bonding interface deficiency.

**Figure 16 sensors-25-00958-f016:**
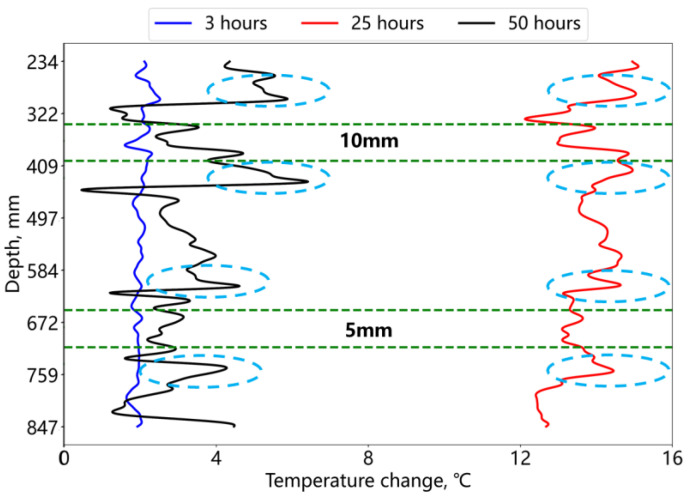
Temperature change at different hydration times for the cement deficiency at the bonding interface.

**Figure 17 sensors-25-00958-f017:**
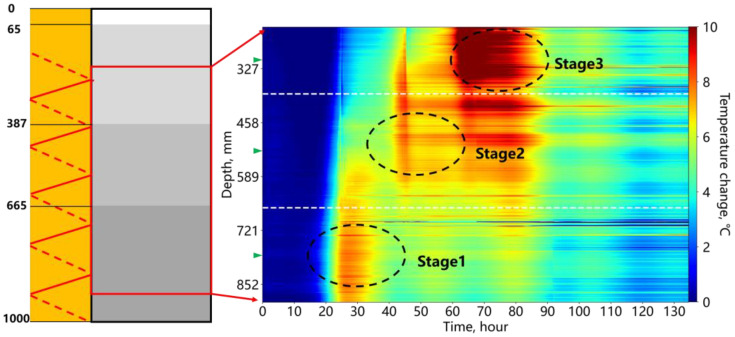
Temperature change during sequential cement injections in the annular space.

**Figure 18 sensors-25-00958-f018:**
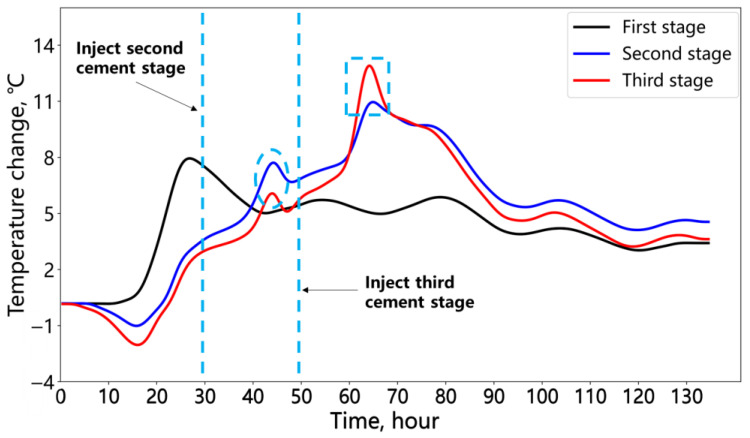
Temperature change of each cement stage in the annular space.

**Figure 19 sensors-25-00958-f019:**
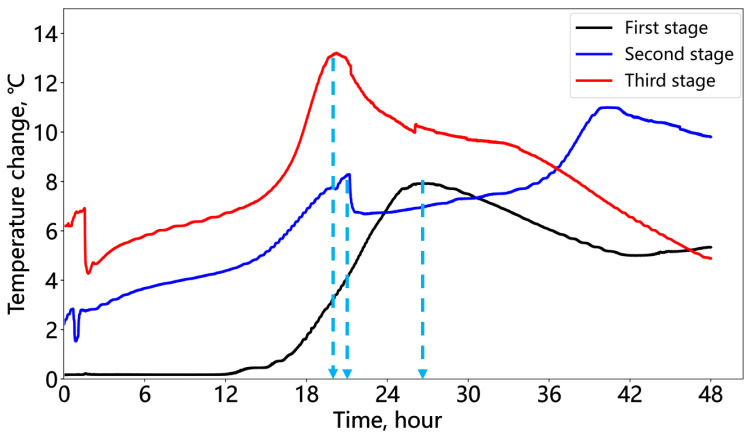
Temperature change in the hydration of the three cement stages over 48 h.

**Table 1 sensors-25-00958-t001:** OSI-D interrogator technical specifications.

Equipment Specifications	Technical Specifications
Measurement Length	20 m
Spatial Resolution	0.64~10.24 mm
Sampling Rate	0~120 hz
Strain Repeatability Accuracy	±1.0~±8.0 με
Temperature Repeatability Accuracy	±0.1~±0.8 °C
Strain Measurement Range	±12,000 με
Temperature Measurement Range	−200~1200 °C
Storage Temperature	0~50 °C
Operating Temperature	10~40 °C
Operating Humidity	<90% RH

## Data Availability

The experimental data in the paper can be obtained directly by contacting the corresponding author, Weibo Sui (suiweibo@cup.edu.cn), or the author, Huan Guo (2024310190@student.cup.edu.cn).

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
