# Peer review of "Experimental Investigation of the Evaluation of the Cement Hydration Process in the Annular Space Using Distributed Fiber Optic Temperature Sensing"

_sensors, 2025, doi:10.3390/s25030958_

Round 1
Reviewer 1 Report
Comments and Suggestions for Authors
The paper "Experimental Investigation on the Evaluation of the Cement
Hydration Process in the Annular Space Using Distributed Fiber Optic Temperature Sensing" is devoted to thermal measurements during cement hydration. The measurements were performed using a fibre optic setup.
In the Reviewer's opinion, the authors must address some of the following issues before acceptance.
- Line 107: The authors use a dynamic distributed fiber optic sensing system for their measurements. However, the information about the device's type, model, and manufacturer is missing. Was it a commercially available device, or did the authors design it for these measurements?
- Line 109: The authors claim that the device's parameters are shown in Fig.1. However, the Reviewer does not see any of the mentioned device's parameters in Fig. 1. This Figure represents only the device's photo and four pictures with the graphic user interface for different measurement modes. Could the authors clarify this?
- Line 138: The authors did not provide relevant information about the high-precision distributed fiber optic temperature and strain measurement system. Was it a commercially available solution or created for this experiment? Also, what kind of optical fibers was used in this system? Was it based on single-mode or multi-mode fibers? Or was it a Bragg grating or something else? This information is crucial. Therefore, the authors should revise this section and provide all the necessary information in the text.
- Line 162: The authors claim that the optical fiber was helically wrapped around the casing at an angle of 20 degrees. The Reviewer would like to know why this angle was selected. Did the authors consider any other angles? Does it have any influence on measurement resolution? The authors should comment on that.
- Line 191: Which Figure (a) do the authors mean? The number of the figure is missing here.
- Fig. 7: Which curve corresponds to which axis? This information is missing.
- Fig. 7: What are the units for the values on the left axis (Fiber response value), and what is the physical meaning of these values?
- Fig. 7: The curves should be shifted so the chart starts from the X-axis's zero value. Also, the zero value should be marked on both vertical axes.
- Line 259: The authors should provide more precise information about fiber optic monitoring. How did the authors collect these spectrograms (presented in Figs. 9, 12, 15). Was it registered by OSI-D interrogator (presented in Fig. 1) or any other device? Please clarify this issue.
- Fig. 17: The red curve shows a rapid increase in the temperature change for ~26 hours. Could the authors explain the cause of this rapid increase?
After addressing these issues, the paper can be reconsidered for publication.
Author Response
Comments 1: [Line 107: The authors use a dynamic distributed fiber optic sensing system for their measurements. However, the information about the device's type, model, and manufacturer is missing. Was it a commercially available device, or did the authors design it for these measurements?]
Response 1: We appreciate your insightful feedback. [The dynamic distributed fiber optic system (OSI-D) is a commercially available device, offering two measurement modes: temperature and strain. The manufacturer is Wuhan Haoheng Technology Company. The company's website is ’https://mega-sense.com/?list_8/36.html’]
Comments 2: [Line 109: The authors claim that the device's parameters are shown in Fig.1. However, the Reviewer does not see any of the mentioned device's parameters in Fig. 1. This Figure represents only the device's photo and four pictures with the graphic user interface for different measurement modes. Could the authors clarify this?]
Response 2: This essential information should be incorporated as a supplement. [Detailed specifications of the distributed fiber optic monitoring system (OSI-D) used in this experiment have been included in the manuscript, covering parameters such as optical fiber measurement length, spatial resolution, measurement accuracy and range for temperature/strain, among others. (Page 3 Line 119-124, Page 4 Table 1)]
Comments 3: [Line 138: The authors did not provide relevant information about the high-precision distributed fiber optic temperature and strain measurement system. Was it a commercially available solution or created for this experiment? Also, what kind of optical fibers was used in this system? Was it based on single-mode or multi-mode fibers? Or was it a Bragg grating or something else? This information is crucial. Therefore, the authors should revise this section and provide all the necessary information in the text.]
Response 3: Thanks for your reminder. [This is a commercially available device, with measurement accuracy suitable for the indoor experiments conducted in this study. The OSI-D system utilizes conventional single-mode optical fibers, with a maximum spatial resolution of 0.64 mm. And the dynamic distributed fiber optic sensing system based on optical frequency domain reflectometry (OFDR) to monitor temperature in real time throughout the entire experimental process (Page 3 Line 119-124)]
Comments 4: [Line 162: The authors claim that the optical fiber was helically wrapped around the casing at an angle of 20 degrees. The Reviewer would like to know why this angle was selected. Did the authors consider any other angles? Does it have any influence on measurement resolution? The authors should comment on that.]
Response 4: Thank you for your constructive feedback. [Experiments with smaller angles were attempted; however, severe bending resulted in excessive optical loss, rendering the measurement data unusable. A winding angle of 20° ensures both precise data measurement and a sufficiently long optical fiber length. (Page 6 Line 188-191)]
Comments 5: [Which Figure (a) do the authors mean? The number of the figure is missing here.]
Response 5: Thank you for your reminder [It’s Figure5 (a) (Page6 Line 202). The titles of all other figures referenced in the manuscript have also been thoroughly checked and revised.]
Comments 6: [ Fig. 7: Which curve corresponds to which axis? This information is missing.]
Response 6: Thank you for your reminder [The left axis represents the values of the dashed line, and the right axis represents the values of the solid line. To distinguish them, we have separated the legends so that the reader can clearly understand.(Page 9 Figure 9 (originally Figure 7)) and Line 268-269)]
Comments 7: [Fig. 7: What are the units for the values on the left axis (Fiber response value), and what is the physical meaning of these values?]
Response 7: [The fiber response value is characterized by the Rayleigh frequency shift, which is measured in Hertz (Hz). The fiber optic measurement of Rayleigh scattering light frequency shift (Hz) has a linear relationship with the temperature change measured by the thermocouple in Figure 9.]
Comments 8: [ The curves should be shifted so the chart starts from the X-axis's zero value. Also, the zero value should be marked on both vertical axes.]
Response 8: Thanks for your advice. We have [made the corresponding revisions, and all similar images in the article have been updated accordingly. (Page 9 Figure 9, Page 10 Figure 10, Page 11 Figure 12, Page 13 Figure 15, Page 15 Figure 18, 19)]
Comments 9: [ Line 259: The authors should provide more precise information about fiber optic monitoring. How did the authors collect these spectrograms (presented in Figs. 9, 12, 15). Was it registered by OSI-D interrogator (presented in Fig. 1) or any other device? Please clarify this issue.]
Response 9: [The OSI-D system measures the Raleigh Frequency shift during the experimental process through a single-mode optical fiber, with the data stored on the computer connected to the OSI-D. The temperature changes monitored by the thermocouple and their correlation with the Rayleigh frequency shift are extracted, and the Rayleigh spectrum is converted into a temperature change spectrum. We utilized Python to read the data, performing necessary downsampling and denoising procedures to obtain the spectrograms. (Page 5 Line 156-164 and Figure 4)]
Comments 10: [Fig. 17: The red curve shows a rapid increase in the temperature change for ~26 hours. Could the authors explain the cause of this rapid increase?]
Response 10: We appreciate your valuable feedback. [The red line represents the temperature change of the third cement stage during the 48-hour hydration period. The sudden increase around 26 hours could be due to a rise in ambient temperature during the day or possibly due to interference between different cement stages. Figure 19 (originally Figure 17) aims to illustrate that the peak temperature during the hydration period occurs at different times for each cement stage. The sudden increase in the red line in the later stage of the hydration period does not significantly impact the interpretation of the figure. We will continue to conduct detailed research to analyze the potential causes.]
Reviewer 2 Report
Comments and Suggestions for Authors
This manuscript reports on the cement hydration monitoring in the annular space through distributed fiber optic temperature sensors based on optical frequency domain reflectometry. The cement sheath and bond interface were monitored during the hydration process under defect conditions, which presence was simulated by means of foams with two different thickness values. Some minor points below should be resolved before the publication of this manuscript.
- Introduction section: other works are reported on the topic, also about hydration monitoring. It would be beneficial to clarify the purpose and the novelty of this work with respect to the previous articles in literature. Moreover, authors state that other works propose FBGs for monitoring the strain response at the cement interface, but “the high cost and difficulties associated with the installation and maintenance of FBGs limit their applicability in field operations”. Since FBGs are low-cost sensors known for their ease of use, please clarify the concept.
- Experiment Setup and Methodology section: Figure 1 reports several screens of the software interface; all the figures in Figure 1 should be numbered as 1a, 1b, etc., and discussed in the text (this is valid for all the figures in the article). Moreover, the last photo in Figure 3 (Figure 3.c) shows cyan and red circles, please explain what they represent. Furthermore, about fiber optics and thermocouples, their relative positioning and locations should be detailed; a photo or schematic reporting of their positions in the structure would be useful.
- Experimental Results and Analysis: Figure 10 “reports the temperature changes at selected points in the cement and deficiency regions”. How were these points selected? Instead, concerning Figure 11, more explanation regarding the fluctuations in correspondence of the deficiency-cement boundaries should be provided.
Author Response
Comments 1: [Introduction section: other works are reported on the topic, also about hydration monitoring. It would be beneficial to clarify the purpose and the novelty of this work with respect to the previous articles in literature. Moreover, authors state that other works propose FBGs for monitoring the strain response at the cement interface, but “the high cost and difficulties associated with the installation and maintenance of FBGs limit their applicability in field operations”. Since FBGs are low-cost sensors known for their ease of use, please clarify the concept.]
Response 1: We appreciate your thoughtful comments, which has significantly enhanced the clarity of our content. [We have added a discussion of the objectives and innovations of this paper in the introduction. Compared to previous work, our approach is innovative in that it utilizes a more accurate distributed fiber optic monitoring system for cement defect identification during hydration period. Additionally, the experimental wellbore dimensions and conditions more closely align with the actual field conditions in oilfields. This allows for a more accurate representation of the data variation amplitude, closely aligning with the actual field conditions. (Page 3 Line 102-111)]
The discussion on fiber Bragg gratings was previously insufficiently rigorous. We have [revised it to: "However, the discontinuity of fiber Bragg gratings under the complex and dynamic downhole conditions may limit their monitoring effectiveness to some extent." (Page 2 Line 93, 94)]
Comments 2: [Experiment Setup and Methodology section: Figure 1 reports several screens of the software interface; all the figures in Figure 1 should be numbered as 1a, 1b, etc., and discussed in the text (this is valid for all the figures in the article). Moreover, the last photo in Figure 3 (Figure 3.c) shows cyan and red circles, please explain what they represent. Furthermore, about fiber optics and thermocouples, their relative positioning and locations should be detailed; a photo or schematic reporting of their positions in the structure would be useful.]
Response 2: Thank you for your reminder. We have [added annotations to Figure 1 and carefully reviewed the other figures in the paper. In Figure 3, the cyan circle indicates a sealing rubber ring on the outer casing, while the red circle marks the cement sheath. To avoid confusion, we have provided these annotations.]
Additionally, [we have labeled the position of the thermocouple, which is located at the midpoint within the cement section and the foam sections. (Page 8 Line 252-255, Page 9 Figure 8)]
Comments 3 [Experimental Results and Analysis: Figure 10 “reports the temperature changes at selected points in the cement and deficiency regions”. How were these points selected? Instead, concerning Figure 11, more explanation regarding the fluctuations in correspondence of the deficiency-cement boundaries should be provided.]
Response 3: We sincerely appreciate your insightful feedback. [We selected the point at the exact midpoint within the foam sections and the cement section to minimize the influence of the boundaries. We have marked the selected points in Figures 11, 14, and 17 using green triangles.]
[The fluctuation observed is due to the different thermal conductivity coefficients of foam and cement. Since the thermal conductivity coefficient of foam is lower, compared with the foam section, the cement section has a faster and larger temperature increase due to the hydration effect. However, heat transfer from the cement section to the foam section was hindered to some extent, thus the temperature near the cement-foam interface higher than the middle section of the cement. (Page 11, Line 320-325)]
Round 2
Reviewer 1 Report
Comments and Suggestions for Authors
After reading the revised version of the manuscript, I can recommend it for publication.
All of my concerns and recommendations have been correctly addressed and explained.
Author Response
Comments 1: [ After reading the revised version of the manuscript, I can recommend it for publication. All of my concerns and recommendations have been correctly addressed and explained.]
Response 1: [We would like to express our sincere gratitude for your thorough review and valuable feedback on our manuscript. Your comments have been instrumental in improving the quality of our paper, and we are pleased that the revised version has adequately addressed all of your concerns and suggestions.]